# Evaluation of the *S*-locus in *Prunus domestica*, characterization, phylogeny and 3D modelling

**Angel Fernandez i Marti** [ID]**[1]\*, Sarah Castro[2], Theodore M. DeJong[2], Richard S. Dodd[1]**

**1** Environmental Science, Policy and Management, University of California, Berkeley, California, United States of America, **2** Plant Science, University of California, Davis, California, United States of America

* afernandezmarti@berkeley.edu

**Data Availability Statement:** Data can be found at NCBI and publicly open. Please find those direct links below https://www.ncbi.nlm.nih.gov/nuccore/MW407936.1 https://www.ncbi.nlm.nih.gov/nuccore/EU113267.1 https://www.ncbi.nlm.nih.

## Abstract

Self-compatibility has become the primary objective of most prune (*Prunus domestica*) breeding programs in order to avoid the problems related to the gametophytic self-incompatibility (GSI) system present in this crop. GSI is typically under the control of a specific locus., known as the *S*-locus., which contains at least two genes. The first gene encodes glycoproteins with RNase activity in the pistils., and the second is an SFB gene expressed in the pollen. There is limited information on genetics of SI/SC in prune and in comparison., with other *Prunus* species, cloning., sequencing and discovery of different *S*-alleles is very scarce. Clear information about *S*-alleles can be used for molecular identification and characterization of the *S*-haplotypes. We determined the *S*-alleles of 36 cultivars and selections using primers that revealed 17 new alleles. In addition, our study describes for the first time the association and design of a molecular marker for self-compatibility in *P. domestica*. Our phylogenetic tree showed that the *S*-alleles are spread across the phylogeny, suggesting that like previous alleles detected in the Rosaceae., they were of trans-specific origin. We provide for the first time 3D models for the *P. domestica* SI RNase alleles as well as in other *Prunus* species, including *P. salicina* (Japanese plum), *P. avium* (cherry), *P. armeniaca* (apricot), *P. cerasifera* and *P. spinosa*.

## Introduction

Self-incompatibility (SI) is an important biological feature of flowering plants, in which pollen-pistil interactions prevent self-fertility and determine the success of mating in out-crossed plants [1]. The mechanism takes on special significance in agricultural fruit and nut crops, where SI reduces the potential for successful fertilization in minimally genetically diverse orchards. For such species, fruit and nut breeders commonly select for self-compatibility that has the added advantage of reduced dependence on insect pollinators [2].

*Prunus*, which belongs to the Rosaceae, is one of the most important genera of fruit and nut trees that includes several commercially important species. Because successful fertilization is critical for fruit and nut production, the genus has been the subject of many studies of the genetic mechanisms of SI and the identification of SI groups that can aid in orchard design. Fruit production is controlled by a gametophytic SI (GSI) system that is regulated by a single

**Funding:** This research was funded in part from
grants to RSD from the California Dried Plum
Board (PN-17-18) and the Innovative Genomics
Institute (Dodd2017RPIL). The funders had no role
in study design, data collection and analysis,
decision to publish, or preparation of the
manuscript. There was no additional external
funding received for this study.

**Competing interests:** The authors have declared
that no competing interests exist.

polymorphic locus containing at least two linked genes; one specifically expressed in the pistil and the other in the pollen [3]. In diploid species, pollen tube growth is arrested in the style whenever the single *S* haplotype expressed in pollen matches one of the two *S* haplotypes expressed in the diploid pistil tissue [4]. The pistil component of SI in the Rosaceae, as in the Solanaceae and Plantaginaceae, is a ribonuclease, *S*-RNase [5, 6], which acts as a cytotoxin in self-pollen tubes, whereas the candidate gene for the pollen component in all *Prunus* species was identified by [7], to be a specific F-box (SFB) gene that is tightly linked to the S-RNase gene [8, 9]. However, the role of SFB proteins is not completely understood.

As would be expected for proper functioning of the GSI system, co-evolution between pollen and stigma parts of the *S*-locus requires mutations in one part to be complemented by mutations in the other [10], leading to expectations of high polymorphism at both loci under frequency dependent selection. Indeed, high S-RNase allelic diversity has been detected for most species studied and high pollen part polymorphism has been reported, for example in *P. dulcis P. dulcis* [7], *P. avium* [11] and in several additional *Prunus* spp. [11]. The pistil *S*-RNases are believed to be ancestral in most eudicots [12], however, diversification of the pollen *S* F-box proteins appears to be more taxon-dependent [7, 13–15], perhaps as a result of the rapid birth/death of F-box genes [16].

Breakdown of the GSI system results in self-compatible (SC) genotypes that are of great interest to the horticultural industry. The presence of haplotypes conferring SC has been studied in diploid *Prunus* species, including almond (*P. dulcis*), Japanese plum (*P. salicina*), sweet cherry (*P. avium*) and apricot (*P. armeniaca*). In most cases SC has been attributed to loss-of-pistil function or to mutations at the SFB gene. For example, in apricot and sweet cherry the SC haplotypes are hypothesized to be a pollen-part mutant in most cases [17–19]. However, the presence of a modifier locus outside the *S* locus has also been reported in sweet cherry, almond and apricot [20–25]. Other putative causes of SC found in Japanese plum and almond have been associated with a low transcriptional level in the $S_e$-RNase [26], or to epigenetic mutations in the $S_f$-RNase [27].

Whereas most work on GSI functioning in *Prunus* has focused on diploid species, several polyploid taxa are important fruit crops, including the tetraploid sour cherry (*P. cerasus*) and the hexaploid prune (*P. domestica*). Functioning of the SI system is likely to be considerably more complicated in polyploids than in diploids and appears to vary among taxonomic groups. For example, polyploidy in the Solanaceae induces a competitive interaction among alleles and the breakdown of the SI interaction with heteroallelic pollen [28]. Competitive interaction was reported by [29] for an SC selection of *P. pseudocerasus* that would be comparable with the Solanaceae. However, in *P. cerasus* [30], have shown that loss of SI is genotype dependent, rather than ploidy dependent, and that, in contrast to the Solananceae, heteroallelic pollen is SI. Detailed crossing studies between *P. cerasus* and the diploid *P. avium* support a one-allele match model, in which pollen is rejected when a functional *S*-haplotype occurs in both diploid pollen and tetraploid stylar tissue and so, breakdown of SI occurs when pollen contains two non-functional *S*-haplotypes that may be due to stylar, or pollen mutations [30, 31].

Among polyploid *Prunus* species, relatively little is known about the GSI system in the hexaploid prune *Prunus domestica* L, including a dearth of information on allelic variation at the two loci within the S haplotype. Lack of this information limits detailed characterization of germplasm, breeding efforts and appropriate orchard design. *S*-genotyping using consensus *S*-RNase primers developed in other *Prunus* species, has successfully identified *S*-RNase alleles [32–36]. However, only three *S*-RNase alleles have been cloned and sequences deposited in the NCBI repository (Sutherland et al., 2008), making genotyping analysis for the remaining alleles very difficult, since these are based on individual interpretations of banding patterns on gels.

In this study we use a multilevel approach that combines traditional field research with molecular gene analysis to understand variation in the *S*-RNase and SFB loci among cultivars and to ascertain the factors affecting reproductive success in *P. domestica*. This integrative approach (pollen tube growth, fruit set evaluation, PCR analysis, genomic DNA cloning/ sequencing and 3D protein modelling) was undertaken in order to identify the SC haplotypes among thirty-six selections developed within the University of California, Davis prune breeding program and to characterize diversity at the *S* locus.

## Material and methods

### Plant material

Pistils and leaves from thirty-six different selections and varieties, including Improved French, Sutter, Muir Beauty and Tulare Giant, were collected from trees growing in the UC Davis prune germplasm collection during three consecutive years (2018, 2019 and 2020). Name of the cultivars and breeding selections are listed in Table 1. Some breeding selections have been derived from crosses among Improved French prune, some others are imported prune clones from France and others are traditional European plum cultivars grown in the US. The UC Davis prune breeding program currently has around 75 selected candidate cultivars for potential release, as well as thousands of seedlings that require evaluation.

### Pollen tube growth and fruit set

Twelve of the 36 selections formed an in vitro study of self-pollination (Table 2). During three consecutive years (2018, 2019 and 2020), at least 12 flower buds a year [37] from each of the 12 seedlings were collected from the field, emasculated and placed in a tray with tap-water, allowing the contact of the flower peduncles with the tray water to prevent dehydration (Fig 1). Anthers were removed from the emasculated flowers and allowed to dry for two days. This pollen was used to self-pollinate the pistils in the tray. Ninety-six hours after pollination, the pistils were placed in tubes containing 5 ml of a 5% solution of $Na_2SO_3$. The samples were maintained at 4°C until observation, when they were stained with 0.1% (v/v) aniline blue in 0.1 N potassium phosphate as a specific stain for callose [38]. This growth was assessed by observation under an Olympus BH2 microscope with UV illumination with a Chiu Technical Corp with a Mercury-100 lamp. A genotype was considered SC when pollen tubes reached the base of at least 8 of the 12 styles, in a minimum of two years. In addition, a branch with a minimum of 100 flowers was bagged before bloom in the field, in order to assess the level of SC by evaluating fruit set in enclosed branches. Sets were calculated 3 months after bagging by counting the total number of fruits and were ranked according to [39], i) less than 2% = SI; ii) between 2% and 5% = low SC; iii) between 5% and 10% = SC and iv) higher than 10% = highly SC.

### Isolation of genomic DNA

In addition to the twelve selections, total DNA from twenty-four more selections was isolated from young leaves using a Qiagen DNA Kit (Qiagen, USA). The quantification and quality evaluation of DNA was performed by a Qubit spectrophotometer (Thermo Fisher Scientific, USA).

### *S*-locus fragment analysis

Pistil *S*-RNase genotyping was performed using four primer pairs that have shown good transferability among other *Prunus* species and have revealed high diversity within the *S*-RNase

**Table 1. Summary of the 36 selections of *P. domestica* used in this study, including their origins and their detected *S*-RNase-genotype.**

| Sample | Name | Pedigree | Detected S-genotype | SC/SI |
|---|---|---|---|---|
| 1 | F3S-5 | Sutter OP | S1/S5/S14 | SI |
| 2 | F11S-38 | French X 6-22-51 | S4/S17 | SC |
| 3 | G5N-35 | French X Sutter | S12/S14/S17 | SC |
| 4 | G45N-35 | D6N103 X Muir Beauty | S8/S14/S16 | SI |
| 5 | G3N-16 | S19-39 X 6-22-51 | S5/S7/S11/S14 | SI |
| 6 | D18S-50 | Imperial X Tulare Giant | S11/S12/S16 | SI |
| 7 | G16N-19 | 3-9E-49 X 4-7E-35 | S11/S14/S17 | SC |
| 8 | G40N-34 | D2N-26 X Muir Beauty | S4/S7 | SI |
| 9 | 2-2E-38RR | Early Tragedy X Primacotes | S5/S7/S9/S16 | SI |
| 10 | D3-39 | French OP | S6/S17 | SC |
| 11 | 3-11E-45RR | French X Tardicotes | S6/S9/S12 | SI |
| 12 | 3-11E-32RR | French X Tardicotes | S2/S3/S4/S12 | SI |
| 13 | 5/1/32 | French OP | S2/S4/S7/S12 | SI |
| 14 | 3-8E-46RR | French X Tragedy | S1/S3/S6/S17 | SC |
| 15 | Tragedy | German Prune X Purple Twain | S7/S12 | SI |
| 16 | Fruit Salad | -- | S3/S6 | SI |
| 17 | Improved French | Agen X seedling of Agen | S10/S12/S17 | SC |
| 18 | Sutter | Sugar X Primacote | S3/S10/S12/S17 | SC |
| 19 | F9N-21 | Tulare Giant X 6-22-52 | S3/S6 | SI |
| 20 | 3-9E-49 | -- | S3/S9/S11/S17 | SC |
| 21 | F11N-34 | 5-3-4 X X 6-19-51 | S4/S6 | SI |
| 22 | G43N-1 | D4N-46 X Muir Beauty | S5/S7/S10/S12 | SI |
| 23 | F13N-23 | French X 6-22-51 | S4/S9/S10 | SI |
| 24 | Jojo | Ortenauer X Stanley | S1/S3/S5/S10 | SI |
| 25 | F2N-10 | 5-19-30 OP | S1/S9/S14/S15 | SI |
| 26 | G5N-58 | French X Sutter | S7/S11/S12 | SI |
| 27 | Tulare Giant | Empress X Primacote | S7/S13/S16 | SI |
| 28 | D6N-103 | French X 5-1-32 | S7/S8/S12 | SI |
| 29 | F9S-33 | Tulare Giant X 6-22-51 | S8/S9/S15 | SI |
| 30 | Muir Beauty | French X Tulare Giant | S10/S17 | SC |
| 31 | F13S-46 | French X 6-22-51 | S4/S10/S11/S15 | SI |
| 32 | G2N-24 | 5-19-39 X 6-15-62 | S4/S15 | SI |
| 33 | 29C | Rootstock | S6/S11/S14/S17 | SC |
| 34 | WS3.1 | -- | S2/S4/S9/S15 | SI |
| 35 | WN16.1 | -- | S7/S8/S12/S15/S16 | SI |
| 36 | H13S-58 | D2N-76 OP | S6/S11/S14/S17 | SC |

Self-compatible (SC) and self-incompatible (SI).

gene [PaConsIF/PaConsIIR and PaConsIF/EMPC5-consRD [10, 37], PruT2/PCER and PruC2/PCER [40, 41]. For the pollen SFB gene, we used the primer combination Fbox5'F/FboxIntronR [42]. All primer information can be found in Table 3.

PCR reactions for the *S*-RNase markers were performed in a volume of 25 μl containing 1x PCR buffer, 1.5 mM MgCl$_2$, 0.2 mM dNTPs, 0.5 μM of each primer and 1 unit of Taq DNA Polymerase. The program consisted of a 2 min denaturation at 94°C, 35 cycles of 1 min at 94°C, 2 min at 50°C and 4 min at 68°C followed by a final extension of 10 min at 68°C.

**Table 2. Data showing fruit set after bagging and pollen tube growth for three years and phenotype for SC/SI from the evaluations in the field and in the lab under microscope observations.**

| Selection | Fruit Set % 2018 | Fruit Set % 2019 | Fruit Set % 2020 | Average. Fruit Set % | Phenotype Fruit set | Number of analyzed pistils | % Pistils with pollen tubes reaching the ovary | Phenotype microscope |
|---|---|---|---|---|---|---|---|---|
| Improved French | 26.7 | 25.2 | 21.1 | 24.3 | Highly SC | 38 | 87 | SC |
| Sutter | 12 | 16 | 14 | 14.0 | Highly SC | 40 | 85 | SC |
| Muir Beauty | 32 | 30.1 | 25.4 | 29.2 | Highly SC | 36 | 91 | SC |
| G5N35 | 19 | 25 | 21 | 21.7 | Highly SC | 35 | 87 | SC |
| G16N19 | 31 | 28 | 23 | 27.3 | Highly SC | 38 | 89 | SC |
| H13S-58 | 7.2 | 6.2 | 5.1 | 6.2 | SC | 36 | 77 | SC |
| G16N19 | 11 | 8.6 | 7 | 8.9 | SC | 37 | 76 | SC |
| D6N-103 | 0.9 | 1.6 | 0 | 0.8 | SI | 37 | 0 | SI |
| D18S-50 | 0 | 0 | 0 | 0.0 | SI | 39 | 0 | SI |
| F3S-5 | 0.7 | 1.5 | 1.2 | 1.1 | SI | 36 | 0 | SI |
| F13S-46 | 0 | 0 | 0 | 0.0 | SI | 36 | 0 | SI |
| Tulare Giant | 1 | 0.6 | 0 | 0.5 | SI | 38 | 0 | SI |

Amplification products were separated on 1% TAE agarose gel in 1X TAE buffer and stained with SYBRGreen and visualized with UV light. A 100 bp DNA ladder was used for fragment size determination.

PCR reaction for the SFB marker 'Fbox5'F/FboxIntronR' was performed with M13 labeled tail in its 5' extremity (CACGACGTTGTAAAACGACA). PCR conditions were as follow, 1X buffer containing 1.5 mM MgCl$_2$., 0.2 mM of each dNTP, 1U Taq polymerase (New England Biolabs), 0.02 μM forward primer with M13 tail in its 5´ extremity, 0.2 μM of M13 Dye (6-Fam) primer and 0.2 μM of reverse primer. The program consisted of 94˚C for 5 min; 30 cycles of 94˚C for 30 s., 56˚C for 45 s, and 72˚C for 45 s; 8 cycles of 94˚C for 30 s, 53˚C for 45 s, and 72˚C for 45 s; and a final elongation step at 72˚C for 10 min. PCR products were detected using an ABI PRISM 3730xl Genetic Analyzer and GeneMapper analysis software (Applied Biosystems, CA, USA) located at the Evolutionary Genetics Lab (UC Berkeley).

## Cloning, sequencing, primer design and analysis of the *S*-RNase alleles

Once the samples were genotyped, the different *S*-RNase alleles were gel-cut, cloned and sequenced. Prior to cloning, the band size corresponding to the target alleles was purified

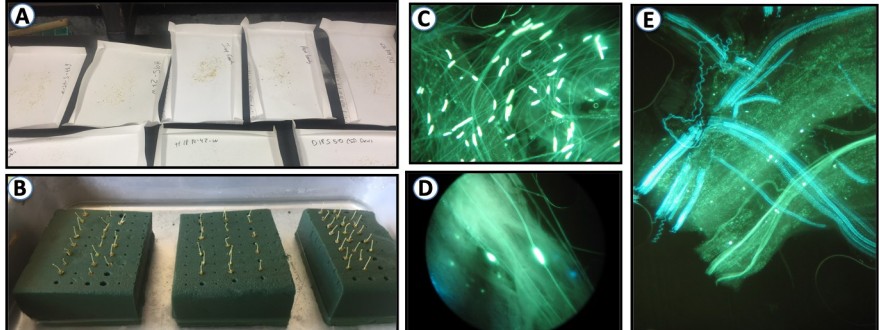

**Fig 1. Pollen tube growth in prune flowers.** Preparation of pollen grains and drying anthers before being used to hand-pollinate the pistils (A), emasculated flowers on wet florist foam (B), pollen grains in self-pollinated flowers of prune germinating at the stigma (C), pollen tubes growing along the style (D), pollen tubes reaching the base of the style (E).

**Table 3. Sequence information of the primers used and designed in this study.**

| Primer name | Sequence 5' 3' | Reference |
|---|---|---|
| PaConsIF | MCTTGTTCTTGSTTTYGCTTT CTTC | [62] |
| PaConsIIR | CAWAACAAARTACCACTTCATGTAAC | [62] |
| EMPC5-consRD | CAAAATACCACTTCATGTAACARC | [33] |
| PruT2 | TSTTSTTGSTTTTGCTTTCTTC | [40] |
| PCER | TGTTTGTTCCATTCGCYTTCCC | [40] |
| PruC2 | CTATGGCCAAGTAATTATTCAAACC | [41] |
| S17-F* | TCTTCCCTTGCTTGGTGTCT | *This study |
| S17-R* | TCCATGTCTGTGTCGGATGT | *This study |
| Fbox5'F | TTKSCHATTRYCAACCKCAAAAG | [42] |
| FboxIntronR | CWGGTAGTCTTDSYAGGATG | [42] |

using the Wizard Plus Miniprep DNA Purification System (Promega, CA, USA) and quantified on 1.5% agarose gel using a standard 1 kb DNA ladder (Invitrogen, CA, USA). The purified PCR products were cloned into the vector pGEM using the Promega Cloning Kit. For each allele, at least three plasmids from different PCR reactions were sequenced from both ends following the methodology described by Fernandez i Marti et al. (2010, 2014). Plasmids were sequenced on an automated sequencer ABI PRISM 3730 Genetic Analyzer (Applied Biosystems, CA). The coding sequences from C1 to C5 of the cloned *S*-alleles, were translated and the deduced amino acid sequences were aligned by the ClustalX method using MegAlign Software (DNASTAR., Madison, WI, USA). *S*-RNase sequences obtained in this work were deposited in GenBank under accession numbers MW407935- MW407936. In order to obtain a complete coverage of the *S*-RNase gene, additional markers were developed by using the primer walking strategy. All primers were designed using the primer3 software [43]. Specific primers for the $S_{17}$ RNase allele were designed (Table 3). The PCR program consisted of an initial denaturation at 94˚C for 2 min, followed by 34 cycles of denaturation at 94˚C for 20 s, 40 s at 57˚C, extension at 72˚C for 1 min, and a final extension at 72˚C for 7 min. PCRs were set up on ice under sterile conditions and the thermocycler was preheated to 94˚C before adding the reactions. PCR products were run on a 1% agarose gel.

## Diversity analysis and evolutionary phylogenetic analysis of *S*-RNase alleles

*P. domestica* is believed to have originated as an interspecific hybrid of a diploid *P. cerasifera* and a tetraploid *P. spinosa*; the latter species may have been an interspecific hybrid of *P. cerasifera* and an unknown Eurasian plum species [44]. The goal of our phylogenetic analysis was to investigate whether the origins of *S*-RNase alleles that we detected in *P. domestica* could be traced to its putative ancestors. Therefore, we obtained available GenBank sequences for 86 published SI alleles from members of subgenus *Prunus sensu* according to [45], see S1 Table. Sequences included *S*-RNase alleles from the putative parents of *P. domestica* (*P. cerasifera* and *P. spinosa*) as well as the diploid plum (*P. salicina*) and seven randomly selected alleles for other *Prunus* species; *P. avium*, *P. dulcis* and *P. armeniaca*. Six *Pyrus communis* L. SI-RNase alleles were used as outgroup. Protein sequences for each species, together with the alleles detected in this study for *P. domestica*, were aligned using ClustalW integrated within the program Geneious. The alignment was visualized and manually refined using Jalview software (www.jalview.org). Phylogenetic analyses were performed using the maximum likelihood method through RAxML 8.0 [46]. The resulting tree was visualized using FigTree v1.4.2.

## Three-dimensional modelling of *S*-RNase proteins

The protein sequence of the *P. domestica* $S_{17}$ RNase (MW407938.1; this work), along with protein sequences obtained from the NCBI database for the SC alleles [*P. dulcis* $S_f$-RNase (QDB64273.1), *P. armeniaca* $S_c$-RNase (ABO34168.1), *P. avium* $S_6$-RNase (AAT72120.1), *P. salicina* $S_e$-RNase (BAF91848.1)] and the SI alleles [*P. domestica* $S_{16}$-RNase (MW407946.1), *P. cerasifera* $S_1$-RNase (AM992048.1) and *P. spinosa* $S_{24}$-RNase (ABV02077.1)] were used for 3D modelling analysis. The modelling procedure began with alignment of the sequences with the related known protein sequence and crystal structure (template) derived from the Protein Data Bank (PDB). The known 3D structure was chosen based on the protein sequence identity, which had to be higher than 35% with our target genes. Sequences were aligned using the deduced protein sequences for these *Prunus* RNase alleles using T-Coffee (EMBL-EBI). The frame of the 3D model was constructed by MODELLER 9v24 [47]. Forty models were constructed for each *S*-RNase. The four models with the lowest value of the Modeller objective function were retrieved for further analysis. Energy function was evaluated through PROSAIIv3 [48].

Stereo-chemical quality and the overall G-factors of the protein models were calculated using PROCHECK [49]. The models with lower numbers of amino acid residues in disallowed regions were selected as the most suitable models. Finally, the molecular graphics of the best models were generated with PYMOL, which visualizes protein structures. Additionally, the 3D models of all S-RNases were compared with that of *Pyrus pyrifolia* $S_3$-RNase (BAA93052.1; [50]), another Rosaceous species.

## Results

### Phenotypic expression

In an attempt to characterize and identify the phenotypes associated with self-compatibility in *P. domestica*, we analyzed the pollen tube behavior in pistils under the microscope after self-pollinations.

Observations of pollen tube growth allowed classifying the phenotype of the selections as SC or SI. Out of the twelve individuals analyzed in the lab, only seven (58.3%) showed pollen tubes at the base of the style (G16N19, G5N35, H13S-58, G16N19, Sutter, Muir Beauty and Improved French). As shown in Table 2, the percentage of pistils with pollen tubes reaching the ovary was between 76% (G16N19) and 91% (Muir Beauty). Consequently, these selections were considered as self-compatible (Fig 1). In the remaining five samples, pollen tube growth was arrested in the middle third of the style of the self-pollinated pistils, and no pollen tubes were observed at the style base or in the ovary., displaying the characteristic arrest of pollen tube growth as the typical SI response (D6N-103, D18S-50, F3S-5, F13S-46 and Tulare Giant). Germinated pollen grains on the stigma were observed in all the selections.

Fruit set after bagging confirmed the SC/SI of all selections previously identified as SC/SI by pollen tube growth. Among the SC selections, five genotypes (G16N19, G5N35, Sutter, Muir Beauty and Improved French) were highly SC according to the criteria of [39] with a fruit set higher than 10%, whereas the selections G16N19 and H13S-58 are considered as SC because fruit sets were between 5% and 10% (Table 2). In addition, sets after bagging confirmed the SI phenotype for the three years of the five individuals previously classified as SI under microscopic observation since their fruit set was lower than 2% in the three years.

### Genotyping, cloning and sequencing analysis of the SI alleles in *P. domestica*

Of all primer combinations tested for the *S*-RNase locus, the primer set PruC2-PCER., which flanks the second and fifth conserved domains of *S*-RNases, was the most successful in all the

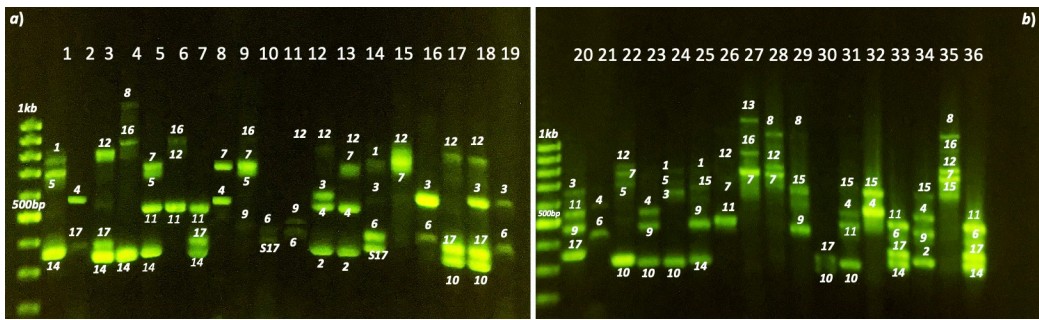

**Fig 2. PCR amplification of 36 dried prune cultivars and selections, obtained with the primers PruC2-PCER.**
Assessment of the alleles was accomplished after individual band isolation, cloning and sequencing.

samples. The number of alleles amplified with the other primer combinations was very low and, in most cases, no more than two alleles were identified for each individual. However, the PruC2-P-CER primer set yielded a multiallelic profile showing from two (Muir Beauty) to five (WN16.1) different fragments per genotype. PCR allele identification revealed seventeen different bands on the gel, including one that was common to all SC phenotypes. This high number of amplified fragments reveals the high genetic diversity for $S$ alleles in prune. Allele sizes ranged from 350 bp ($S_{10}$) to 1350 bp ($S_{13}$). The most common alleles were $S_{17}$ that was common to all SC phenotypes and $S_4$, $S_{14}$, $S_{12}$ and $S_{11}$ which were observed in eight, or nine different selections (Fig 2). By contrast, allele $S_{13}$ was only detected in one selection (Tulare Giant). A maximum of five alleles were found in one genotype (WN16.1). The average number of alleles per individual was three (Table 1).

We cloned and sequenced all bands amplified by primers PruC2-PCER to identify new alleles. Once all gel-band alleles were cloned and sequenced, a "blastn" analysis was conducted in order to find nucleotide sequence similarities with other $S$-alleles from *Prunus* species present in NCBI database. For all 17 alleles, we found high sequence similarities (≥95%) with S-RNase alleles from a range of species (Table 4).

PCR amplification of the intron of the pollen SFB revealed a total of fourteen different alleles within the thirty-five *P. domestica* genotypes for both primer combinations. The size obtained for Fbox5'F/Fbox–IntronR ranged between 192 bp and 375 bp, with the allele size 235 bp being the most common (found in 9 samples). By contrast, the SFB alleles with a size of 201, 207 and 240 were only detected in one selection.

## A $S$-RNase allele common to self-compatible phenotypes of *P. domestica*

As described earlier, the $S_{17}$ allele was present and common to all self-compatible phenotypes but was not detected in self-incompatible selections (Table 1). Sequences of three different cloned colonies for this band in multiple individuals of Improved French and Sutter showed a nucleotide match of 100%. Within the genus *Prunus*, the $S_{17}$ allele from *P. domestica* showed 94% similarity with the $S_5$-RNase allele described for *P. virginiana* (JQ627793.1). We designed specific primers to amplify this allele. The new molecular marker was then used to detect presence/absence in the remainder of the selections. The primer pair produced a single band, with a size of 205 bp (Fig 3), in all genotypes that were classified as SC according to our phenotypic analyses. As expected, this new marker did not amplify any band in the SI genotypes.

## Phylogenetic tree analysis and genetic relationship among the $S$-RNase alleles within the Rosaceae

We inferred evolutionary relationships among the $S$-RNase alleles of *P. domestica*, and among other *Prunus* species (*P. salicina*, *P. avium*, *P. dulcis*, *P. armeniaca*, *P. spinosa* and *P. cerasifera*),

**Table 4. Size in base pairs (bp) of the new *S* alleles discovered in this study for RNase and SFB genes and sequence similarities with *S*-RNase alleles from a range of species in the genus *Prunus*.**

| Allele this study | Orthologous allele | Species with orthologous gene | % similarity | PRUC2-PCER (bp) | Fbox5'F-FboxIntronR (bp) |
|---|---|---|---|---|---|
| $S_1$ | $S_{17}$RNase | *P. salicina* | 100 | 780 | 375 |
| $S_2$ | $S_{27}$RNase | *P. salicina* | 100 | 380 | n.a |
| $S_3$ | $S_6$RNase | *P. domestica* | 96 | 615 | n.a |
| $S_4$ | $S_1$RNase | *P. persica* | 98 | 585 | 230 |
| $S_5$ | $S_5$RNase | *P. domestica* | 100 | 695 | 235 |
| $S_6$ | $S_6$RNase | *P. domestica* | 100 | 310 | 274 |
| $S_7$ | $S_4$RNase | *P. cerasifera* | 99 | 745 | 201 |
| $S_8$ | Unidentified | *P. cerasifera* | 99 | 1250 | n.a |
| $S_9$ | $S_9$RNase | *P. domestica* | 100 | 470 | 365 |
| $S_{10}$ | $S_{10}$RNase | *P. cerasifera* | 97 | 350 | 196 |
| $S_{11}$ | $S_{24}$RNase | *P. salicina* | 98 | 525 | 240 |
| $S_{12}$ | $S_{18}$RNase | *P. virginiana* | 97 | 805 | 192 |
| $S_{13}$ | $S_{31}$RNase | *P. speciosa* | 95 | 1350 | 225 |
| $S_{14}$ | $S_{12}$RNase | *P. virginiana* | 95 | 365 | 215 |
| $S_{15}$ | $S_{24}$RNase | *P. spinosa* | 98 | 710 | 207 |
| $S_{16}$ | $S_8$RNase | *P. spinosa* | 95 | 890 | 242 |
| $S_{17}$ | $S_5$RNase | *P. virginiana* | 94 | 395 | 300 |

n.a, no amplification.

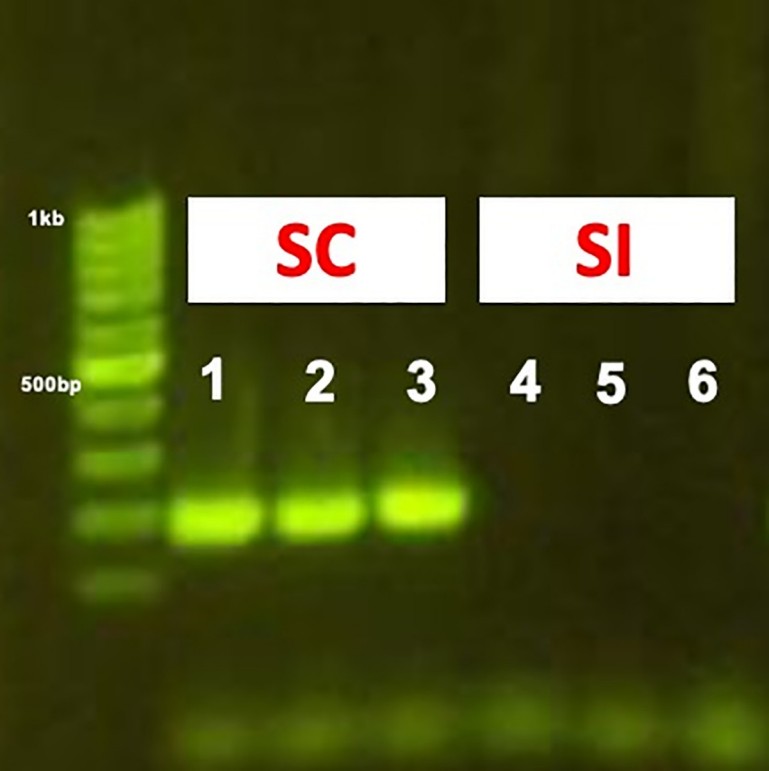

**Fig 3. Example of efficacy of the new $S_{17}$ marker tested in three SC and three SI samples.** Improved French (1), Sutter (2), Muir Beauty (3), Tulare Giant (4), 18S-50 (5) and 6N-103 (6).

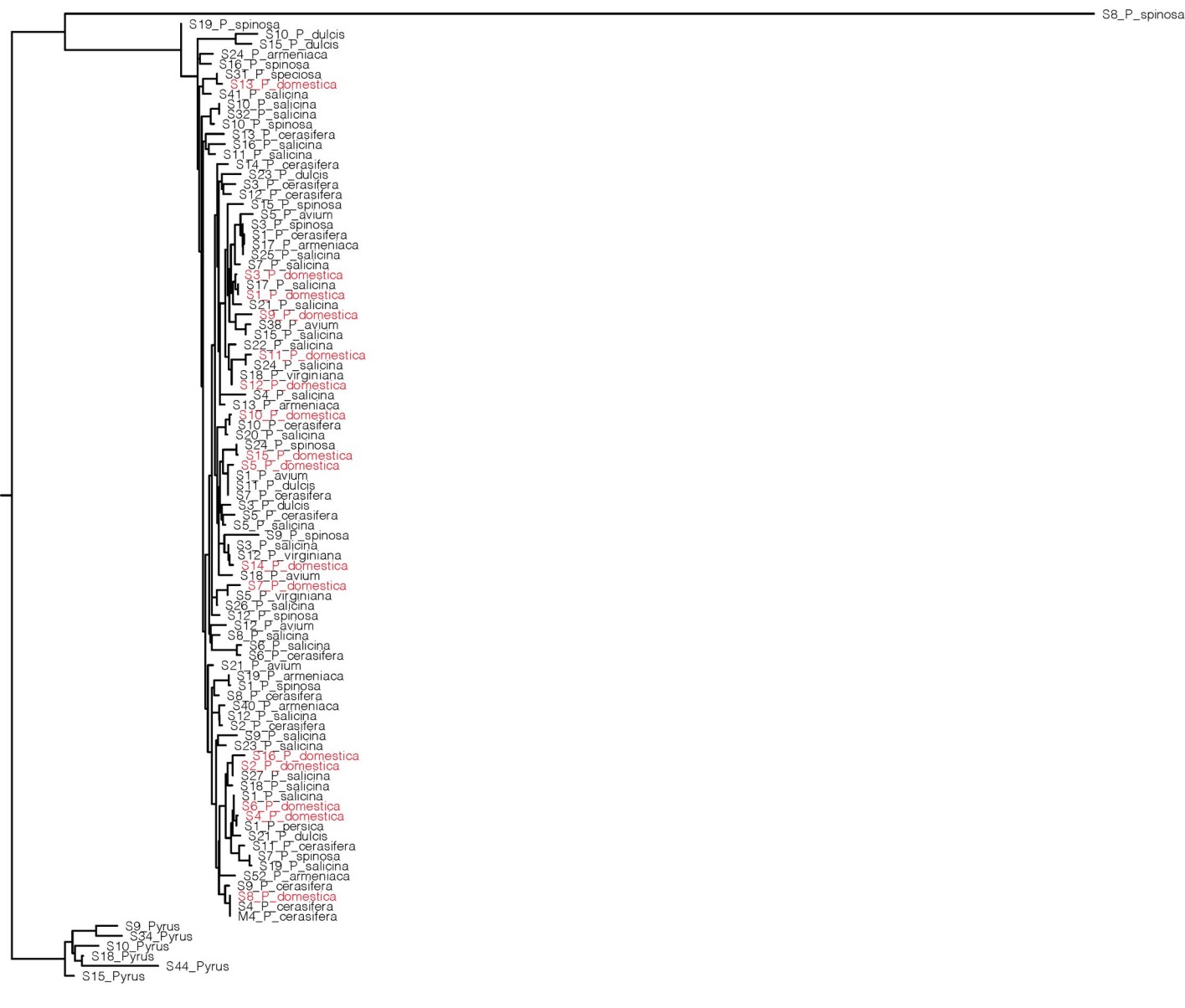

**Fig 4. Phylogenetic relationships among *S*-RNase alleles from Prunus species including, the 17 alleles identified in this study.** Pyrus was used as an outgroup.

by constructing a ML phylogenetic tree of amino acid sequences (Fig 4). The *P. domestica S*-RNase alleles were widely dispersed in the ML phylogenetic tree, supporting previous reports of trans-specific origins. There was no evidence that *P. domestica S*-RNase-alleles clustered more frequently with *S*-RNase-alleles of putative hybrid parents, *P. spinosa* and *P. cerasifera*, than with alleles of any of the other *Prunus* species.

### 3D modelling of *S*-RNase proteins

We constructed 3-D models of the SC-RNase alleles of *P. dulcis*, *P. armeniaca*, *P. avium* and *P. salicina* and compared these with the SI alleles of *P. domestica*, *P. cerasifera* and *P. spinosa* $S_{24}$-

RNase as well as the *P. domestica* $S_{17}$ based on related known 3D structures derived from the Protein Database (PDB). The best template model chosen was the RNase MC1 mutant with accession 1J1G, with an identity between this template and the SC *P. domestica* $S_f$-RNase of 41% (E-value = 3e$^{-28}$). The similarity to the SI *P. domestica* $S$-RNase was 40% and E-value = 9e$^{-32}$. The MC1 protein sequence has 42% identity and E-value = 4e$^{-33}$. The similarity to the $S_e$-RNase, $S_6$-RNase and $S_c$-RNase was 40%., 38% and 37.2%, respectively (E-value = 3e$^{-28}$, E-value = 9e$^{-32}$ and E-value = 1e$^{-26}$). The similarity and identity to *P. cerasifera* $S$-RNase and the *P. spinosa* $S_{24}$-RNase were 39% and 39.5%, with E-value = 1e$^{-29}$ and E-value = 3e$^{-37}$.

Protein structures include secondary structural elements, with α-helices from the core regions of the molecule connected by loop regions (β-sheets) on the protein surface. Our $S$-RNases belonged to the α and β class, with six α helices and six β sheets connected by loops. The folding topologies of the main chains, found in the models analyzed here, were very similar to the topologies of the RNase T$_2$ family of enzymes. The MC1 protein sequence we used as a template has eight α-helices and eight β sheets, whereas protein structures generated for the SC *P. salicina*, *P. avium*, *P. spinosa*, *P. armeniaca* and the $S_{17}$ *P. domestica* alleles had 7 α-helices and between 2 and 6 β sheets (six for *P. salicina* and *P. spinosa*; four for *P. domestica* and *P. avium*; and two for *P. armeniaca*). The protein structure generated in *P. cerasifera*, SI *P. domestica* and SC *P. dulcis* $S_f$-RNase was 6 α-helices and 6 β sheets, with an overall molecular dimension for each $S$-RNase of approximately 40 Å x 50 Å x 30 Å.

Ramachandran plot statistics showed that 95.3% of amino acid residues were positioned in the favored regions. Structures that place 95–97% or more of the amino acid residues in the favored positions, are considered to be reliable in modelling experiments. Thus, these results indicate that our models were optimal.

When all $S$-RNases were super positioned, the *P. domestica* $S_{17}$-RNase structure contained an additional extended loop, which was either not present or was much shorter than in the remaining SI $S$-RNases. When we aligned the amino acid sequences of the *P. domestica* $S_{17}$-RNase and the SI *P. domestica* $S$-RNase, we found that this long loop was associated with 17 amino acid residues (magenta loop in Fig 5) located between the conserved domains RC4 and C5 and corresponding to the sequence KRHSAQTKSGPKPLLLH. Zooming in on this 3D model structure revealed a very small loop found in the SI RNase allele in *P. domestica*, which had only 7 amino acids in that region of the gene (cyan loop; Fig 5).

## Discussion

In this work, we report phenotypic expression of SI/SC and the isolation and characterization of novel $S$-RNase alleles in thirty-five genotypes of the hexaploid *P. domestica*. Also, we report on allelic variation within the SFB pollen part gene in the same selections of *P. domestica*.

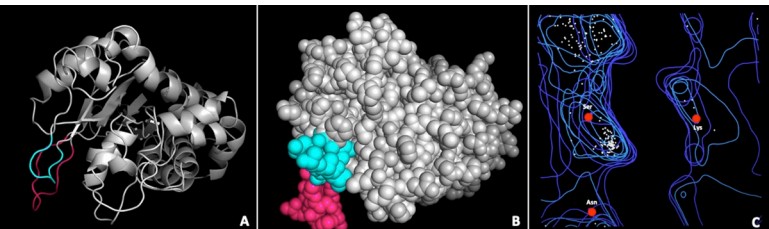

**Fig 5.** Stereo representation (a), ribbon diagram (b) and Ramachandran plot (c) of modeled structure of *P. domestica* $S_{17}$-RNase (magenta), and *P. domestica* SI-RNase (cyan), showing secondary structural elements and surfaces.

## Self-compatibility in *P. domestica*

To determine the status of self (in)compatibility in prunes, bagged branches and self-pollinations in the lab were conducted for three years. The phenotypes of the twelve selections were consistent using the two approaches. Pollen tube growth has been considered a clear indication of the compatibility of any pollination as it is independent of the environment where the study is done as in all cases the results have been unequivocal [51]. However, fruit set evaluation in the field is subject to many environmental hazards in spite of being the most natural approach to the real self-compatibility level of any genotype [2]. To our knowledge, this research represents the first study in *P. domestica* to evaluate self-compatibility for three years.

Pollen tube growth observations were robust during the three consecutive years, however, the overall number of fruit set was slightly lower in 2020 than in 2018 and 2019. It has been reported in other species such as almond and tomato, that environment may strongly affect fruit set [2, 52]. Although we observed a reduced fruit set in 2020, the sets obtained showed the ability of the flowers to set fruit with their own pollen. The studies conducted in the field show the most reliable response since they reflect the natural conditions for pollination.

## Detection of *S*-RNase alleles

Self-incompatibility has been extensively studied in the most important diploid *Prunus* species, including Japanese plum, cherry, apricot and almond [19, 21, 22, 26, 41]. For initial *S* allele genotyping, we used SFB primers and consensus primers that amplified the *S*-RNase gene. These molecular markers of *S*-RNase and SFB loci have commonly been utilized to study the genetics of self-incompatibility in other *Prunus* species. For the *S*-RNase pistil component, we obtained reliable amplifications only with the PruC2-PCER primer combination that had been developed in sweet cherry [41]. Similar results were described by [36], where, out of all the primers tested, the PruC2-PCER primer combination was the most successful. Some of the other conserved primers were developed in almond, subgenus *Amygdalus*, suggesting possible low transferability to the subgenus *Prunus*. Although [33] suggested that primers spanning the 2$^{nd}$ intron, such as PruC2, were not very informative in *P. domestica* due to a narrow range of amplification fragments, we were able to amplify up to five different fragments per sample. Our results agreed with previous *S*-genotyping analyses which revealed a high number of amplified fragments in prune by using the same primer combination [34].

We compared our *S*-genotyping results with phenotypical evaluations in the field and observations of pollen tube growth in pistils after self-pollinations under the microscope. In total, we detected seventeen different *S*-RNase alleles, one of which ($S_{17}$) was common and exclusive to all SC phenotypes among the thirty-five prune cultivars. The alleles we detected included the RNase of the three haplotypes ($S_5$, $S_6$ and $S_9$) reported by [33]. The most frequent alleles among our tested selections were $S_4$, $S_{11}$, $S_{12}$, $S_{14}$ and $S_{17}$ which, based on the blast analysis, had high similarities with alleles described in *P. salicina*, *P. persica* and *P. virginiana*. Although the similarity of our alleles was high when compared to other species (between 95% and 98%), we observed several indels and nucleotide differences. Thus, our results suggest that they can be considered as new alleles within the Rosaceae. Only the three alleles reported in *P. domestica* by [33] plus the alleles $S_1$ and $S_2$ that were 100% identical to $S_{17}$RNase and $S_{27}$RNase in *P. salicina*, have been previously published in the NCBI GenBank.

## Self-compatibility in *P. domestica*

Approximately one third of the total cultivars and selections we analyzed (31.5%) were self-compatible and could be detected with a new molecular designed to amplify the $S_{17}$ allele (F11S-38, G5N-35, G16N-19, D3-39, 3-8E-46RR, Improved French, Sutter, F13S-46, Muir

Beauty, 29C, and H13S-58). Many of these selections have their origin in Improved French, which is well known to be a SC variety and so these selections could have inherited the $S_{17}$ allele in common. Self (in)compatibility in polyploid species is likely to be considerably more complex than in diploids. The breakdown of SI in polyploid members of the Solanaceae results from competitive interaction among S alleles, such that heteroallelic pollen grains (those containing two different pollen S alleles) are compatible with stylar tissue regardless of its genetic composition [28]. However, a relationship between polyploidy and increased SC is not likely to be a general rule. Indeed, genotypic variation in SI/SC expression in the tetraploid *P cerasus* provides strong evidence that SC is not determined by ploidy in this species [30]. Our data with the hexaploid *P. domestica*, showing a mix of SI and SC genotypes, supports this latter observation that polyploidy *per se* is not the cause of SC in these polyploid *Prunus* species. The breakdown of SI in Improved French has not been investigated and could have several origins, including stylar *S* RNase, and pollen SFB mutations, or factors external to the *S*-locus. We found a 300bp fragment of the SFB intron that was associated with the SC selections and correlated with the $S_{17}$ RNase allele. Therefore, we cannot speculate whether the loss of SI among our selections of *P. domestica* may be caused by stylar or pollen mutations. Mutations in the SFB gene appear to be most common in SC diploid *Prunus* species e.g. *P. armeniaca* [17, 19], *P. mume* [53] and *P. avium* [18] whereas both stylar and pollen part mutations have been reported to cause breakdown of SI in the polyploid *P. cerasus* [54, 55]. SC may also be determined by a factor external to the *S* locus. This was suggested by [20] for the variety Cristobalina of *P. avium* and recent molecular studies have detected a putative modifier region. Interestingly, in Cristobalina, self-fertilized pollen tube growth is slower than cross-fertilized, suggesting a form of partial SC [20]. A modifier gene mediating the pollen part conferring SC has also been detected in *Prunus armeniaca* [24, 25] and in *Prunus dulcis*, where some genotypes lacking the SC allele had a self-compatible phenotype in the field [21].

## *S*-RNase relationships among *Prunus* species including putative parents of hexaploid *P. domestica*

The sequence differences that we detected at the *S*-RNase locus agree with the hypothesis that new S-alleles may be generated by the accumulation of point mutations within the female component [56]. These genetic variations can arise from gene mutations, or from genetic recombination near introns, as suggested by [57], leading to alterations in gene activity or protein function. We found high sequence similarity ($\geq 95\%$) for S-RNase alleles with a range of species in the genus *Prunus*, suggesting orthologous alleles. The ML phylogenetic tree showed no evidence that alleles detected in *P. domestica* were more closely associated with those of the putative parents, *P. spinosa* and *P cerasifera* than with any other of the *Prunus* species studied. The *P. domestica* alleles were spread across the phylogeny, consistent with trans-specific origins described by [58] and others [54, 55], although, as pointed out by [59], the allele origins in *Prunus* are not as ancient as in some other groups, such as the Solanaceae.

## 3-D structure of *S*-RNase alleles

We used PYMOL to visualize the 3-D structure of amino acid residues within the *S*-RNase gene. It has been proposed that loops in 3D structures serve to link α-helices and β-strands, and that longer loops could be susceptible to proteolytic degradation. The general functional differences among 3D almond SC and SI RNases involved an extended loop in the $S_f$-RNase [60, 61]. Furthermore, the main structural difference found in the three-dimensional models generated in this study between the SC and SI *P. domestica* proteins resided in the presence and length of the loop within the conserved domains RC4 and C5. When we analyzed the

modeled SC proteins for the $S_e$-RNase of *P. salicina*, $S_{6'}$-RNase of *P. avium* and $S_c$-RNase of *P. armeniaca*, the three 3D structures also presented the same loop pattern and were similar in length to the SC *P. domestica*. The protein structure of the SI RNases of *P. spinosa* and *P. cerasifera* either lacked this loop., or the loop was shorter than that of the SC alleles in *P. avium*, *P. armeniaca*, *P. dulcis* or *P. domestica*. Variation in the number, lengths, and positions of α-helices, β sheets and loops seems to contribute to functional differences among *S*-RNases in *Prunus*. Although it is unknown if this long loop has a direct effect on autogamy in plants, it is interesting that all the SC *S*-RNases in *Prunus* share this unique feature.

## Conclusion

This study describes for the first time a large number of S-RNase alleles in the hexaploid *P. domestica*. *S*-genotyping, cloning and sequencing allowed us to identify and characterize 17 *S*-RNase alleles, of which 13 are considered new and can be used for orchard design and selection of parental genotypes in prune breeding programs. In addition, our study describes for the first time the association and design of a molecular marker for self-compatibility in *P. domestica*. Our phylogenetic tree showed that the *S*-alleles, are spread across the phylogeny, indicating trans-specific origin. Three-dimensional *S*-RNase structures were analyzed for the first time in plum, cherry, apricot as well as in *P. cerasifera* and *P. spinosa*.

## Supporting information

**S1 Table. *S* alleles, species name and accession number for each sequence obtained from NCBI used for the construction of the phylogenetic tree in Fig 4.**
(DOCX)

**S1 Raw images.**
(PDF)

## Acknowledgments

AFM wants to dedicate this article to Dr. Socias i Company, who passed away last November 2020. Dr. Socias was one of the pioneers to discover the trait of SI in fruit trees, almost 50 years ago, and it is from him that AFM inherited this passion for working in tree breeding and genetics.

## Author Contributions

**Conceptualization:** Angel Fernandez i Marti.

**Formal analysis:** Angel Fernandez i Marti.

**Funding acquisition:** Theodore M. DeJong, Richard S. Dodd.

**Investigation:** Angel Fernandez i Marti, Richard S. Dodd.

**Resources:** Sarah Castro, Theodore M. DeJong, Richard S. Dodd.

**Supervision:** Richard S. Dodd.

**Validation:** Angel Fernandez i Marti, Richard S. Dodd.

**Writing – original draft:** Angel Fernandez i Marti, Richard S. Dodd.

**Writing – review & editing:** Angel Fernandez i Marti, Theodore M. DeJong, Richard S. Dodd.

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
