## [Decision Letter · Decision Letter 0]

23 Mar 2021

PONE-D-21-08430

Evaluation of the S-locus in Prunus domestica, characterization, phylogeny and 3D modelling

PLOS ONE

Dear Dr. Fernandez i Marti,

Thank you for submitting your manuscript to PLOS ONE. After careful consideration, we feel that it has merit but does not fully meet PLOS ONE’s publication criteria as it currently stands. Therefore, we invite you to submit a revised version of the manuscript that addresses the points raised during the review process.

We look forward to receiving your revised manuscript.

Kind regards,

Jen-Tsung Chen, Ph.D.

Academic Editor

PLOS ONE

Journal Requirements:

"This research was supported, in part, by the California Prune Board. "

Reviewers' comments:

Reviewer's Responses to Questions

**Comments to the Author**

1. Is the manuscript technically sound, and do the data support the conclusions?

Reviewer #1: Yes

Reviewer #2: Yes

2. Has the statistical analysis been performed appropriately and rigorously? 

Reviewer #1: Yes

Reviewer #2: Yes

3. Have the authors made all data underlying the findings in their manuscript fully available?

Reviewer #1: Yes

Reviewer #2: No

4. Is the manuscript presented in an intelligible fashion and written in standard English?

Reviewer #1: Yes

Reviewer #2: Yes

5. Review Comments to the Author

Reviewer #1: In the current study, the authors have characterized the S-locus in Prunus domestica. Further, its phylogeny and 3D modelling have also been carried out. Overall, I do not see any major issue with the experimentation. However, only a few points can be addressed before further consideration.

Line 47-52, please support your statements with vital reference(s).

Line 771/Table 1, remove “S” from the end of the caption, and define SC/SI in the table caption or as a footnote.

Currently, I could not find any information for MW407935- MW407936. Please follow the journal policy regarding data availability.

Line 307, could you please provide us with microscopic images (within text or suppl. figure) of the pollen tube behavior in pistils.

Figure 3 quality is very poor (96 dpi), and the text is not clearly visible. Please provide an image with a higher resolution.

It would be nice to add the Ramachandran plot in figure 4 together with 3D models.

During the screening, I found that several sentences are very similar to previous publications. The authors must rephrase the plagiarized text. Please see the attached report.

Reviewer #2: In this study, one of the important aspects of self-incompatible plant breeding has been investigated. Although the results are small, but enough innovation is provided.

Some comments and suggestions:

- Add references to the first paragraph of introduction.

- Introduction is so long, so I suggest to remove/merge some sentences.

- Discussion needs to improve.

6. PLOS authors have the option to publish the peer review history of their article (what does this mean?). If published, this will include your full peer review and any attached files.

Reviewer #1: No

Reviewer #2: No

---

## [Author Response · Author response to Decision Letter 0]

20 Apr 2021

Dear Editor:

I am re-submitting the manuscript “Evaluation of the S-locus in Prunus domestica, characterization, phylogeny and 3D modelling” by Sarah Castro, Ted DeJong, Richard Dodd and myself for publication in Plos One.

We have followed all the reviewer’s suggestions and attached the new documents as requested by the journal.

“This research was funded in part from grants to RSD from the California Dried Plum Board (PN-17-18) and the Innovative Genomics Institute (Dodd2017RPIL). The funders had no role in study design, data collection and analysis, decision to publish, or preparation of the manuscript. There was no additional external funding received for this study”.

We hope that you will find the study interesting and within the scope of Plos One. 

Thanking you very much for your attention.

RESPONSE TO REVIEWERS

Reviewer #1: In the current study, the authors have characterized the S-locus in Prunus domestica. Further, its phylogeny and 3D modelling have also been carried out. Overall, I do not see any major issue with the experimentation. However, only a few points can be addressed before further consideration.

Line 47-52, please support your statements with vital reference(s).

Done. We have added two new references.

Line 771/Table 1, remove “S” from the end of the caption, and define SC/SI in the table caption or as a footnote.

Done. We have added the definition of SC/SI

Currently, I could not find any information for MW407935- MW407936. Please follow the journal policy regarding data availability.

Done. We have request to NCBI to make our sequences publicly available and now are open to everyone.

Line 307, could you please provide us with microscopic images (within text or suppl. figure) of the pollen tube behavior in pistils.

Included a new picture (Picture 1) and labelled the rest following the new order. 

Figure 3 quality is very poor (96 dpi), and the text is not clearly visible. Please provide an image with a higher resolution.

Done. Picture improved at 300dpi and tiff format

It would be nice to add the Ramachandran plot in figure 4 together with 3D models.

Done. We have added a new Ramachandran plot in figure 5(c).

During the screening, I found that several sentences are very similar to previous publications. The authors must rephrase the plagiarized text. Please see the attached report.

Done, we have removed and edited new sentences.

Reviewer #2: In this study, one of the important aspects of self-incompatible plant breeding has been investigated. Although the results are small, but enough innovation is provided.

Some comments and suggestions:

- Add references to the first paragraph of introduction.

Done. We have added two new references.

- Introduction is so long, so I suggest to remove/merge some sentences.

Done. We have e followed the reviewer advice and have shortened the introduction

- Discussion needs to improve.

The reviewer requested improvement to the Discussion without any further details. We have taken a close look and made a few changes including some reorganization that we feel makes the manuscript stronger and clearer

---

## [Decision Letter · Decision Letter 1]

26 Apr 2021

Evaluation of the S-locus in Prunus domestica, characterization, phylogeny and 3D modelling

PONE-D-21-08430R1

Dear Dr. Fernandez i Marti,

We’re pleased to inform you that your manuscript has been judged scientifically suitable for publication and will be formally accepted for publication once it meets all outstanding technical requirements.

Kind regards,

Jen-Tsung Chen, Ph.D.

Academic Editor

PLOS ONE

Additional Editor Comments (optional):

Reviewers' comments:

Reviewer's Responses to Questions

**Comments to the Author**

1. If the authors have adequately addressed your comments raised in a previous round of review and you feel that this manuscript is now acceptable for publication, you may indicate that here to bypass the “Comments to the Author” section, enter your conflict of interest statement in the “Confidential to Editor” section, and submit your "Accept" recommendation.

Reviewer #1: All comments have been addressed

2. Is the manuscript technically sound, and do the data support the conclusions?

Reviewer #1: Yes

3. Has the statistical analysis been performed appropriately and rigorously? 

Reviewer #1: Yes

4. Have the authors made all data underlying the findings in their manuscript fully available?

Reviewer #1: Yes

5. Is the manuscript presented in an intelligible fashion and written in standard English?

Reviewer #1: Yes

6. Review Comments to the Author

Reviewer #1: The authors have addressed all the reviewer comments. Notably, the MS has been improved and ready for acceptance. Thus, I am endorsing the current version for publication in Plos One.

Congratulations and best wishes for future publications!

7. PLOS authors have the option to publish the peer review history of their article (what does this mean?). If published, this will include your full peer review and any attached files.

Reviewer #1: No

---

## [Editor Report · Acceptance letter]

4 May 2021

PONE-D-21-08430R1 

Evaluation of the S-locus in *Prunus domestica*, characterization, phylogeny and 3D modelling 

Dear Dr. Fernandez i Marti:

I'm pleased to inform you that your manuscript has been deemed suitable for publication in PLOS ONE. Congratulations! Your manuscript is now with our production department. 

Kind regards, 

on behalf of

Dr. Jen-Tsung Chen 

Academic Editor

PLOS ONE